# Pathophysiological Role of Neutrophil Extracellular Traps in Diet-Induced Obesity and Metabolic Syndrome in Animal Models

**DOI:** 10.3390/nu17020241

**Published:** 2025-01-10

**Authors:** Andrej Feješ, Katarína Šebeková, Veronika Borbélyová

**Affiliations:** Institute of Molecular Biomedicine, Medical Faculty, Comenius University, 83303 Bratislava, Slovakia; andrejfejesb@gmail.com (A.F.); veronika.borbelyova@imbm.sk (V.B.)

**Keywords:** NETosis, diet-induced obesity, mice, PAD4, citH3, MPO

## Abstract

The global pandemic of obesity poses a serious health, social, and economic burden. Patients living with obesity are at an increased risk of developing noncommunicable diseases or to die prematurely. Obesity is a state of chronic low-grade inflammation. Neutrophils are first to be recruited to sites of inflammation, where they contribute to host defense via phagocytosis, degranulation, and extrusion of neutrophil extracellular traps (NETs). NETs are web-like DNA structures of nuclear or mitochondrial DNA associated with cytosolic antimicrobial proteins. The primary function of NETosis is preventing the dissemination of pathogens. However, neutrophils may occasionally misidentify host molecules as danger-associated molecular patterns, triggering NET formation. This can lead to further recruitment of neutrophils, resulting in propagation and a vicious cycle of persistent systemic inflammation. This scenario may occur when neutrophils infiltrate expanded obese adipose tissue. Thus, NETosis is implicated in the pathophysiology of autoimmune and metabolic disorders, including obesity. This review explores the role of NETosis in obesity and two obesity-associated conditions—hypertension and liver steatosis. With the rising prevalence of obesity driving research into its pathophysiology, particularly through diet-induced obesity models in rodents, we discuss insights gained from both human and animal studies. Additionally, we highlight the potential offered by rodent models and the opportunities presented by genetically modified mouse strains for advancing our understanding of obesity-related inflammation.

## 1. Introduction and Background

Obesity is a chronic multifactorial disease defined by excessive fat deposits in fat tissues, or ectopically—in organs typically containing small amounts of lipids, such as the liver, epicardium, pancreas, or skeletal muscle. The prevalence of obesity rises worldwide: in adults, an increase from 42% in 2020 to over 54% by 2035 is expected. According to present trends, by 2035, two children in every five globally will be overweight or obese [1]. The rocketing incidence of obesity is a global health crisis with far-reaching impacts on individual health and quality of life, national economies, and social systems.

However, obesity prevalence varies significantly across regions due to socioeconomic, cultural, genetic, and environmental differences. In high-income countries, the prevalence is high, generally contributed to by the combination of a sedentary lifestyle and consumption of high-calorie processed foods. In middle-income countries, the prevalence of obesity is rising, mainly in urban areas, where lifestyle is favoring the transition to consumption of a Western-type diet and sugar-sweetened beverages in combination with reduced physical activity. In contrast, rural areas often face undernutrition. Low-income countries confront persisting undernutrition and food insecurity in the majority of the population, with the increasing prevalence of obesity in mainly affluent subgroups. Interestingly, the Pacific Islands and the Gulf States record the highest prevalence of obesity due to genetic predisposition, cultural norms favoring large body size, and a fondness for imported calorie-dense foods. These differences necessitate region-specific, tailored health strategies addressing the unique drivers with tailored interventions and specific prevention, addressing social determinants but also integrated approaches and global collaboration [1,2,3].

Health consequences of obesity may arise directly from the increased mass of fat tissue, such as mobility disability or osteoarthritis, or indirectly, mediated via several substances secreted from the enlarged (particularly) visceral fat tissue depots, exerting endocrine effects, especially on the cardiovascular system. Thus, obesity is associated with an increased risk for cardiovascular diseases (CVDs)—as well as other conditions like chronic kidney disease, endocrine or psychiatric disorders, and some types of cancer [4,5]—and, mainly at high body mass index (BMI) levels, with an increased risk of cardiovascular or all-cause mortality [6].

### 1.1. Obesity-Associated Low-Grade Inflammation

Obesity is a state of chronic low-grade inflammation, often termed meta-inflammation, as it affects all metabolic organs. A key factor initiating this state is hypertrophy of visceral adipocytes, which outgrow their vessel supply and become hypoxic, stressed, and may undergo necrosis. Hypoxia leads to stromal response via hypoxia-inducible factor-1α, inducing the expression of pro-inflammatory adipokines and cytokines. While subcutaneous fat secretes biologically active molecules into the systemic circulation, pro-inflammatory molecules from visceral fat leak directly into the portal circulation, stimulating lipogenesis in the liver, ultimately resulting in fatty liver, insulin resistance (IR), and cardiometabolic dyslipidemia [7]. Pro-inflammatory adipokines released from stressed hypoxic and necrotic adipocytes also lead to the recruitment and activation of leukocytes [8,9].

Substances secreted by adipose tissue do not unequivocally induce cardiovascular dysfunction. For example, adiponectin, apelin, interleukin (IL)-10, and IL-4 are anti-inflammatory and reduce oxidative stress. Thus, at the time of diagnosis, not all obese subjects are clinically (metabolically) ill. This state, often called “metabolically healthy obesity”, is more frequent in young persons and females. It is considered a transient phenotype, reflecting an early stage in the development of obesity before the overt manifestation of cardiometabolic disorders progressing to non-communicable diseases [10]. With progressing adipocyte hypertrophy, the profile of biological factors secreted by fat cells into the extracellular space changes into pro-inflammatory, pro-oxidant, and pro-atherogenic [6].

### 1.2. The Role of Neutrophils in Obesity-Associated Meta-Inflammation

Neutrophils play a critical role in innate immunity by eliminating intracellular pathogens through microbicidal mechanisms that can be oxygen-dependent and oxygen-independent [11,12]. The oxygen-dependent microbicidal activity of neutrophils involves the production of reactive oxygen species (ROS) via the respiratory burst process mediated by an enzyme complex—NADPH oxidase [13]. Activation of NADPH oxidase leads to the rapid production of superoxide anions (O_2_^−^), which in turn are converted to hydrogen peroxide (H_2_O_2_) by superoxide dismutase (SOD). Additionally, H_2_O_2_ might be converted by myeloperoxidase (MPO) to hypochlorous acid (HOCl), a potent antimicrobial agent. These ROS are highly toxic to bacteria and other pathogens, leading to their destruction. On the other hand, the oxygen-independent microbicidal activity of neutrophils relies on antimicrobial proteins (lysozyme, lactoferrin, defensins, and cathelicidins) and enzymes (elastase, cathepsin G, proteinase 3) stored in neutrophil granules [14]. These components are released into the phagolysosome, creating a hostile environment for the engulfed pathogens, ultimately leading to their death.

Several key functions of neutrophils contribute to the chronic low-grade inflammation and metabolic disturbances characteristic of obesity. Neutrophils are among the first immune cells to infiltrate expanding (visceral) adipose tissue in obesity, as stressed adipose tissue releases C–X–C motif chemokine ligands (CXCL), which attract neutrophils. Once in adipose tissue, neutrophils can recruit and activate other immune cells by releasing CXCL2. Affecting T-cell activation, they further amplify inflammatory responses. Neutrophils release pro-inflammatory mediators such as ROS and cytokines, triggering inflammation by recruiting macrophages. This results in converting M2 macrophages with anti-inflammatory phenotype to pro-inflammatory M1 macrophages. Lipids, such as free fatty acids, adipokines, and pro-inflammatory cytokines, mediate this shift [8,15]. Except for influencing the differentiation of monocytes into pro-inflammatory M1 macrophages and releasing ROS, neutrophils promote systemic inflammation and drive insulin resistance via increasing endoplasmatic reticulum stress in adipocytes and initiation of inflammation in adipose tissue by secretion of tumor necrosis factor-α (TNF-α) and IL-6 [16]. Since both dysfunctional adipocytes and activated leukocytes produce pro-inflammatory molecules, the continuous low-grade inflammation of obesity becomes self-sustained [17,18].

Recent knowledge suggests that a different mechanism also plays a role in obesity-associated inflammation and complications—neutrophil extracellular traps (NETs) formation, termed NETosis. Originally, it was described as “a form of innate response that binds microorganisms, prevents them from spreading, and ensures a high local concentration of antimicrobial agents to degrade virulence factors and kill bacteria” [19]. The specific association of NETs with obesity and its related comorbidities came later when studies began linking NET formation with sterile inflammation and metabolic disorders [20,21].

## 2. Formation of Neutrophil Extracellular Traps

Neutrophils use different host defense strategies: phagocytosis, degranulation, chemical signaling, and releasing neutrophil extracellular traps (NETs) [19,22]. The primary mechanism of their phagocytic and antimicrobial function is a release of granular substances, such as metalloproteinases (MMP-8, MMP-9), MPO, neutrophil gelatinase-associated lipocalin (NGAL), and neutrophil elastase (NE) [23,24,25]. To form NETs, neutrophils also release their DNA into the extracellular space.

### 2.1. Pathogen- and Danger-Associated Molecular Patterns

Formation of NETs—extracellular, web-like structures containing granule and cytosolic proteins on a scaffold of decondensed chromatin—is a pathway to trap, neutralize, and kill pathogens. Pathogen-associated molecular patterns (PAMPS) are specific arrangements of key molecules found on or within pathogens and typically conserved across their entire classes. Since they are not present in the host organism, the immune system recognizes PAMPs as foreign. Thus, PAMPs play a crucial role in the early detection and initiation of the immune response [26]. Neutrophils recognize lipopolysaccharides (LPS), peptidoglycans, flagellins, unmethylated CpG islands of DNA, or viral DNA/RNA as PAMPs through pattern recognition receptors (PRR) [27]. Toll-like receptors (TLRs) on the surface of neutrophils recognize pathogen components; those localized intracellularly detect double- and single-stranded RNA or unmethylated CpG DNA. When PRRs bind to PAMPs, they trigger an immune response, producing pro-inflammatory cytokines, recruiting other immune cells, and initiating processes like phagocytosis to eliminate the pathogen [28,29,30].

NETosis is an essential but double-edged sword in the immune defense system. While NETs play a critical role in trapping and neutralizing pathogens, they can also be triggered by endogenous molecules. These molecules include specific components of NETs themselves or substances released from apoptotic, pyroptotic, or necrotic cells. Examples of such endogenous triggers, collectively termed damage-associated molecular patterns (DAMPs), include nuclear DNA (ncDNA), mitochondrial DNA (mtDNA), histones, high-mobility group box protein-1 (HMGB-1), S100 proteins, and heat shock proteins. DAMPs activate the innate immune system by interacting with PRPs, such as TLRs or receptors for advanced glycation end-products (RAGE), promoting increased cytokine production and, thus, further recruitment and activation of neutrophils [31,32]. This may create a vicious cycle of persistent inflammation, potentially resulting in organ damage, systemic inflammation, organ failure, or even death [32,33,34,35].

### 2.2. Forms of NETosis

NETosis occurs in distinct forms depending on the stimuli and mechanisms involved. The main forms are suicidal, vital, and mitochondrial NETosis (Figure 1). Except for mitochondrial NETosis, most extracellular DNA (ecDNA) in NETs originates from the nucleus (ncDNA). Suicidal NETosis is more prominent in chronic inflammation and infections, while vital and mitochondrial NETosis is often involved in acute responses and sterile inflammation. Excessive or dysregulated NETosis is believed to play a pathogenetic role in inflammatory diseases like obesity, diabetes, atherosclerosis, and autoimmune disorders [19,32,36,37,38,39,40,41,42,43,44,45].

#### 2.2.1. Suicidal NETosis

Suicidal NETosis involves the neutrophil undergoing cell death to release NETs. It is an ROS-dependent process, typically taking 2 to 4 h to complete. Neutrophil activation is triggered by intense stimuli, such as strong pathogens, microbial products, inflammatory cytokines, or DAMPs, via TLRs or other PRRs. These stimuli induce the extracellular signal-regulated kinase (ERK)-dependent assembly of the nicotinamide adenine dinucleotide phosphate (NADPH) oxidase, which generates ROS in the cytoplasm to release granule proteins and drives chromatin decondensation in the nucleus. Granule proteins such as neutrophil elastase (NE) and MPO translocate to the nucleus and assist in the degradation of histones and breaking down nuclear components. Peptidyl-arginine deiminase 4 (PAD4) catalyzes the hydrolytic deamination of arginine residues in histones of chromatin to citrulline and promotes loosening chromatin structure. NADPH oxidase-derived ROS are essential for initiating the pathways that activate gasdermin D (GSDMD), which induces pore formation in granule membranes (to enhance the release of proteases), nuclear envelop (aiding the mixing of nuclear chromatin with cytoplasmic antimicrobial proteins), and plasma membrane (facilitating the final step of membrane rupture and the release of NETs). The cell membrane eventually breaks, releasing NETs (DNA–protein complexes) into the extracellular space, resulting in neutrophil death. Alongside, antimicrobial proteins like defensins, cathelicidins, and lysozymes are released, enhancing the antimicrobial effect [46,47,48,49].

#### 2.2.2. Vital NETosis

Vital NETosis is a form of NET formation that preserves neutrophil functionality while releasing NETs to trap pathogens. This process occurs within 5 to 60 min of stimulation by bacteria, bacterial products, complement proteins, or TLR4-dependent platelet–neutrophil interactions. Vital NETosis can be ROS-independent. Chromatin decondensates within a nuclear envelope but is released from the nucleus in vesicles along with antimicrobial proteins. Vesicles fused with the plasma membrane and chromatin are expelled into the extracellular space through vesicular transport, forming NETs, while the cytoplasm and granules remain intact. The neutrophil that committed vital NETosis remains viable and functional, capable of performing other immune functions, such as migration, degranulation, and elimination of bacteria by phagocytosis [47,50,51,52,53,54].

#### 2.2.3. Mitochondrial NETosis

Mitochondrial NETosis occurs rapidly, within 15 to 20 min. It involves a release of mtDNA along with mitochondrial and cytoplasmic proteins into the extracellular space, where it forms web-like structures. The release of mtDNA is triggered by mitochondria-generated ROS and is thought to involve autophagy-like mechanisms. It is considered a type of vital NETosis, allowing neutrophils to conserve nuclear integrity and maintain other functions. Mitochondrial NETs may have distinct functions, especially in autoimmune diseases [55,56,57,58].

Obesity is characterized by mitochondrial dysfunction resulting from low-grade inflammation and associated oxidative stress [59]. It is speculated that mitochondrial NETosis may play a key role in driving the low-grade sterile inflammation associated with obesity, which is triggered by neutrophils infiltrating the visceral adipose tissue (VAT). Mitochondria possess their own hereditary DNA. MtDNA has a circular structure, resembling bacterial DNA in form and size, and mitochondrial ribosomes, enzymes, and transport systems exhibit characteristics akin to bacteria. Mitochondria’s evolutionary bacterial ancestry and shared structural features with prokaryotes explain why signaling pathways activated by external (pathogenic) and internal (endogenous) stimuli overlap. MtDNA (through TLR9) and formyl peptides (via the formyl peptide receptor-1) can act as DAMPs, activating pathways promoting inflammation across various tissues, similar to those triggered by bacterial PAMPs [60,61,62].

### 2.3. Metabolic Pathways in NETs Formation

NETosis is a regulated process requiring energy, fueled via different metabolic pathways. Since the number of mitochondria in neutrophils is lower compared to other immune cells, glycolysis (via the Krebs cycle, pentose phosphate pathway, glycogenolysis, glutaminolysis, and fatty acid β-oxidation) is utilized to fulfill the energetic requirements [63,64,65,66]. Depending on the type of inducer, NETosis may be ROS-dependent or ROS-independent [67]. Induction by pro-inflammatory cytokines, nitric oxide, oxidized LDL (oxLDL), or phorbol 12-myristate 13-acetate (PMA) stimulates NETosis via NADPH oxidase (NOX2)-mediated oxidative burst; thus, it is termed ROS-dependent. Inducers such as uric acid, soluble immune complexes, calcium ionophores, and diverse microorganisms that do not require NOX2 activation are termed NOX-independent. NET formation via suicidal NETosis is ROS-dependent [68]. Thus, lacking NADPH oxidase leads to diminished NET formation [69]. To generate NETs, mitochondria may compensate for ROS production, and mitochondrial NETosis is fueled by mitochondria-derived ROS (mitoROS) [39].

Initiation of NETosis by both NOX-dependent and -independent processes requires glycolysis. During NETosis, the glycolysis rate increases, pyruvate levels decrease, and increased lactate formation suggests the involvement of the Warburg effect. Lactic acid produced by bacteria or lactate released under hypoxic inflammatory conditions by expanded adipose tissue also induces NETs formation [70]. Hyperglycemia may either enhance the production of NETs or, via induction of spontaneous neutrophil activation, impair the ability to respond to LPS, reducing ROS production and phagocytosis [71,72,73]. In addition, the non-esterified fatty acids and oxidized HDL or oxLDL promote NETosis [74,75,76].

### 2.4. Interplay of Neutrophils and NETs with Other Immune Cells

NETosis, per se, plays a critical role in systemic inflammation. Still, interactions of NETs with other immune cells may modulate the inflammatory response, e.g., NETs may activate macrophages to produce pro-inflammatory cytokines. Macrophages may promote NET resolution via phagocytosis, but excessive NETosis may overwhelm this clearance mechanism, leading to chronic inflammation [77,78]. NETs may affect T cell function directly, by presenting antigens, or indirectly, via the production of cytokines or the cytotoxicity of NET components, such as histones. Thus, in certain conditions, NETs can be cytotoxic to T cells, limiting their activity, while in autoimmune diseases, they drive autoreactive T cell responses [79,80]. Similarly, by presenting autoantigens, NETs drive B cell activation and autoantibody production, particularly in autoimmune diseases. At the same time, cytokines released during NETosis may promote B cell survival and differentiation into antibody-producing plasma cells [81,82]. Platelets may induce NETosis, and vice versa—creating a feedback loop that eventually amplifies thromboinflammation [83,84]. Last, NET components act as chemoattractants for other neutrophils, amplifying the inflammatory cascade.

### 2.5. Sex Differences

In the general population of adults (n > 6400, age: >55 years), no sex differences in plasma concentration of NETs were reported [85]. Neither a study in healthy children (10 ± 5 years, n = 16), young (29 ± 4 years, n = 11), old (73 ± 4 years, n = 16), and very old (88 ± 6 years, n = 11) adults [86], nor that in a cohort of patients with severe obesity (formerly termed morbid obesity) (age range: 20–72 years, n = 73) indicated differences between males and females [87]. However, healthy male adolescents (n = 550) presented with higher plasma cfDNA concentrations and a higher number of ncDNA and mtDNA genome equivalents (GE) than females (n = 699) [88]. Plasma concentrations of advanced oxidation protein products (AOPPs, representing mainly MPO-modified albumin) were directly and significantly associated with mtDNA GE in both sexes. Whether AOPPs are markers of mitochondria-induced NETosis remains unclear. However, these data raise the question of whether, in adults, existing comorbidities mask the sex differences.

### 2.6. Age Differences

In the mentioned large study on the general population of adults, plasma concentrations of NETs (quantified as MPO–DNA complexes) showed a weak inverse relationship with age [85]. On the other hand, in a cohort of healthy subjects and patients living with severe obesity (n = 128, age range: 20–75 years), or patients with obesity per se, no significant relationship between plasma concentrations of MPO–DNA complexes and age was revealed [87]. In another cross-sectional study, old and very old subjects displayed significantly higher plasma concentrations of MPO–DNA complexes and of NET markers (e.g., plasma MPO, NE, citrullinated histone H3—citH3) compared with young adults [86]. Since the plasma activity of deoxyribonuclease (DNase) was similar between the groups, higher concentration of NETs in the elderly was attributed to their higher resistance to degradation by DNase I, probably due to a higher degree of oxidatively modified DNA, as suggested by higher levels of 8-hydroxy-2′-deoxyguanosine in NETs [86].

### 2.7. Priming of NETosis by Sterile Stimuli

Compared to young adults, only neutrophils of elderly subjects reacted by exaggerated NETs formation if co-incubated with mitochondria, a stimulus mimicking the formation of NETs under sterile conditions. Moreover, plasma from elderly individuals induced NETosis in neutrophils of healthy donors more potently than plasma from healthy young adults or children [86]. Further addition of mitochondria boosted NETosis more powerfully in old than in young individuals. This priming by residual NETs was mediated through TLR9 signaling. In a different study, neutrophils isolated immediately after trauma injury from elderly patients released, after stimulation by mtDNA, fewer NETs than those of younger patients [89]. This discrepancy may reflect the fact that mitochondria, in contrast to mtDNA, may activate neutrophils via several other metabolic pathways except TLR9, such as formyl peptide receptors 1 and 2 or TLR4. The size of mtDNA vs. mitochondria may also matter since neutrophils tend to respond with NET formation more likely when interacting with large pathogens [90,91].

### 2.8. Association of NETosis with Inflammatory Status

Supplementation of plasma of healthy donors with recombinant interleukin IL-1β, IL-8, or TNF-α primed healthy neutrophils to form NETs via activation of NADPH oxidase and MPO [92]. In the abovementioned study [86], plasma concentrations of these pro-inflammatory cytokines did not differ significantly between the age groups. Still, a proportion of old subjects displayed concentrations of IL-6, interferon (INF)-α2, and monocyte chemoattractant protein (MCP)-1 indicative of mild low-grade inflammation. MCP-1 promoted NET formation in neutrophils of healthy donors [93]. Transcriptomic data support the role of IL-6 in triggering NETosis [94].

### 2.9. Effects of Temperature on NETs Formation

In vitro and ex vivo data (using neutrophils or whole blood from healthy donors, respectively) show that NETs formation and their clearance by DNase rise by increasing temperature in the range of 36 °C to 41 °C, whereas this response is attenuated by hypothermia and hyperthermia (at 35 °C and 42 °C, respectively) [95]. These data are not only clinically relevant regarding, e.g., sepsis, but point to the need to maintain physiological temperature during ex vivo experiments.

## 3. Obesity-Associated Neutrophilia

### 3.1. Human Studies

Asymptomatic individuals living with obesity (n = 50) presented with mild neutrophilia accompanied by higher levels of acute-phase reactants [96]. Patients living with severe obesity (n = 27) displayed higher levels of low-density neutrophils characterized by a pro-inflammatory gene signature associated with inflammation, neutrophil activation, and immunosuppressive function compared with their lean peers (n = 20). In a longitudinal study, improvements in anthropometric indicators, metabolic syndrome (MS) components, and inflammatory markers and a decline in low-density neutrophil counts were observed after bariatric surgery (n = 37) [97]. The visceral adipose tissue (VAT) of patients living with obesity (n = 82) also contained a higher proportion of neutrophils than that of lean individuals (n = 14). The adiponectin expression was lower, and that of leptin and pro-inflammatory genes was higher in the VAT of obese patients than in lean controls. Insulin resistance, leptin expression, and pro-inflammatory genes directly correlated with VAT neutrophil abundance. Moreover, the genetic signature of adipose tissue and peripheral blood neutrophils differed, with the former displaying a higher expression of pro-inflammatory and anti-apoptotic genes. These data suggest that the rise in adipose tissue neutrophils observed in human obesity is not merely a consequence of increased migration of peripheral blood neutrophils. Instead, adipose tissue neutrophils appear to possess functional characteristics distinguishing them from circulating neutrophils [98].

#### Summary

Although studies associating neutrophil counts with BMI are scarce, they concordantly show that patients with obesity, particularly those with severe obesity, present with mild but significant neutrophilia. A single study showed that neutrophil abundance in VAT is higher in patients living with obesity than in lean subjects and that bariatric surgery-induced reduction of BMI results in a decline in low-density neutrophil counts. These findings are highly interesting, but further studies to reexamine these tendencies are desired due to the small sample size of the studies and the majority of probands being females.

### 3.2. Experimental Studies

Studies using animal models of obesity indirectly support the potential role of NETosis in obesity-associated metabolic disturbances. In mice, a high-fat diet (HFD) induced infiltration of adipose tissue with neutrophils accompanied by increased expression of NE was associated with impaired glucose tolerance. This impairment was not present in NE knock-out mice. In wild-type animals, NE administration boosted glucose intolerance, while it improved with the administration of NE inhibitors [99]. Another study showed that in HFD-induced obesity in mice, the gut microbiome affects neutrophil infiltration-associated inflammation of visceral adipose tissue and insulin sensitivity [98].

## 4. Neutrophil Extracellular Traps in Obesity

### 4.1. Human Studies

Patients living with severe obesity (mean BMI: 45.5 kg/m^2^) displayed significantly higher plasma concentrations of NETs (MPO–DNA complexes) than lean, healthy controls. Moreover, concentrations of the MPO–DNA complexes before bariatric surgery were associated with cardiometabolic risk factors, such as proxy measures of obesity, blood pressure (BP), and markers of glucose metabolism, but not with lipids profile. Albeit all patients (n = 73) responded to bariatric surgery with a decrease in body weight and improvement of cardiometabolic risk factors and markers, plasma MPO–DNA levels decreased only in a subgroup of patients (n = 40). Those in whom circulating NET levels increased post-operationally had a higher incidence of thromboembolic events in medical history (4/33 vs. 0/40) [87]. Thus, surgery-induced reduction of adipose tissue volume does not unequivocally correct NET’s dysregulated production. Whether persisting high levels of NETs predict a higher risk of cardiovascular events remains unclear.

Individuals subjected to bariatric surgery (n = 10) displayed a higher proportion of circulating neutrophils and higher plasma and visceral adipose tissue levels of MPO–DNA complexes than their eutrophic counterparts (n = 10). Bioinformatics and proteomics analyses suggested that IL-8 (a macrophage-derived protein triggering rapid migration of neutrophils), heat shock protein 90 (HSP90, molecular chaperon potentially promoting inflammation and thus activating NETs signaling pathways in obesity), and the E1 heat shock protein family HSPE1 (the cochaperone for HSP 60 inhibiting the inflammatory response) could modulate NETs formation in obesity [100].

Another study found that neutrophils from patients with severe obesity, when stimulated ex vivo, produce more ROS and release higher levels of pro-inflammatory cytokines but form fewer NETs than neutrophils from healthy, eutrophic controls. Among 18 subjects in each group, 17 were females. Weight loss after gastric banding was associated with a decline in ROS production and cytokine release and an increase in NET production. Thus, metabolic surgery-induced weight loss improved the hyperreactive pro-inflammatory phenotype of neutrophils and restored the diminished ability of neutrophils to form NETs in response to stimuli, which could ultimately be reflected in better ability to combat infection [101].

Obesity increases the risk of developing asthma and complicates asthma management, as obese asthmatics have worse lung function compared to non-obese ones, in part probably due to the increased airway neutrophilia [102]. Thus, Williams et al. focused on the role of NETs in the obese asthma phenotype. While the concentration of ecDNA in sputa was similar between non-obese (n = 65) and obese (n = 69) patients, NET abundance assessed as DNA–NE complexes was significantly higher in obese patients, and it was associated with the percentage of sputum neutrophils and poorer lung function [103].

#### Summary

Obesity is a state of chronic low-grade inflammation characterized, among others, by increased NET formation. Reduced NET formation after bariatric surgery-induced weight loss points to a potential resolution of this inflammatory state. This could contribute to improved immune function; a lower risk of obesity-related comorbidities; and, thus, better cardiovascular health. Understanding how fat mass reduction leads to decreased NET formation opens possibilities for therapeutic interventions that mimic the effects of weight loss or specifically target NET formation to restore immune balance and treat chronic inflammatory conditions or autoimmune diseases. The persistence of NETs after bariatric surgery probably negates the mentioned benefits, potentially leading to prolonged inflammation, hindering the improvement of insulin resistance, and increasing the risk of cardiovascular complications. The persistence of NETs may also be associated with surgical or medical complications necessitating additional monitoring and management strategies. Aimed longitudinal studies are needed to confirm or refute this scenario. Whether higher NET counts in subjects living with obesity reflect enhanced mitochondrial NETosis remains to be elucidated.

### 4.2. Animal Models

#### 4.2.1. Rodent Models of Obesity

The global pandemic of obesity boosted research on pathophysiology and potential new treatment modalities of diet-induced obesity. Relatively long-life spans, small size, short reproductive cycles, and high reproductive output make small rodents a choice for preclinical studies. However, it is tricky to fatten up rodents. When sugar-sweetened beverages are co-administered with a standard or a simple Western-type diet, animals compensate for the additional calories from the soft drink and consume less of the solid food [104,105,106]. Thus, high-caloric, obesogenic diets, such as HFD or HFD combined with high fructose or high sugar diets (HF-HSD), are the most widely used [107]. In HFDs, 40–60% of calories come from fats, generally lard, or palm oil. Hence, these diets do not mirror or correspond to the obesogenic diets consumed by humans. On the other hand, HFDs are available from commercial suppliers, enabling reproducibility and comparability between studies. Data suggest that rather than HFD type, the crucial element to achieving dysregulated metabolism and desired visceral fat accumulation is the duration of HFD consumption. A reasonable alternative to HFD is a cafeteria diet consisting of unhealthy, highly palatable processed foods with attractive odors and tastes [108], reflecting human consumption patterns of obesogenic Western-type diets. The cafeteria diet concurrently enables the individual choice of food preferences, which play crucial roles in human obesity development. The variety and novelty of food items served in the cafeteria diet induce hedonic feeding; voluntary hyperphagia; and, subsequently and rapidly, obesity [109,110,111]. Recently, a standardized cafeteria diet feeding protocol was proposed [112]. The globalization of the food market and the existence of international trade food chains allow for reproducibility between studies.

Although HFDs lead to obesity and related health problems, they lack the diversity and appeal of actual human diets, making cafeteria diets more representative for studying obesogenic effects in rodent models. In modeling dietary obesity-induced MS-like traits such as visceral obesity, insulin resistance, and low-grade inflammation, the cafeteria diet outperforms the HFD. At the same time, the HFD is more efficient in inducing atherogenic dyslipidemia [109,113,114,115]. However, the two diets affect some cardiometabolic risk markers differently: the cafeteria diet is more potent in the induction of vascular dysfunction and changes in gut microbiota, while HFD consumption is associated with hypercoagulability [115,116]. Moreover, similarly to humans, the responses to and health consequences of obesogenic diets differ between male and female rodents [117,118,119].

To our knowledge, data on NET formation in the model of cafeteria diet-induced obesity is not available. However, we bring this model to attention since the cafeteria diet is rich in advanced glycation end-products (AGEs)—brownish substances formed on thermally processed foods, rendering them attractive in taste and odor. Dietary AGEs are partially absorbed into circulation. Their interaction with the cell-surface pattern recognition receptor RAGE induces downstream pro-inflammatory and pro-atherogenic pathways and the formation of ROS [120]. Functional RAGE is present on the plasma membrane of neutrophils, and engagement of RAGE impairs neutrophil functions [121,122].

#### 4.2.2. NETosis in Dietary-Induced Obesity Models

Compared with eutrophic male mice, those with HFD-induced obesity displayed higher levels of cfDNA/MPO complexes in white adipose tissue but not in plasma. Intravital microscopy confirmed the colocalization of neutrophils with DNA in epidydimal adipose tissue. In HFD-consuming mice, adiposity was associated with a dysmetabolic phenotype characterized by higher plasma total cholesterol, triacylglycerols, glucose, and ionized calcium levels; lower HDL-C concentration; lower activity of SOD; and higher activity of catalase [100]. Compared with lean counterparts, obese mice (both the wild-type with HFD-induced obesity or genetically obese leptin-deficient ob/ob mice), either healthy or endotoxemic, displayed higher glycemia, cholesterolemia, alanine aminotransferase activity, liver weight, and altered liver morphology. Counts of NETs in liver sinusoids were lower than in eutrophic animals. The diminished in vivo ability of neutrophils of obese mice to form NETs spontaneously stemmed from an altered platelets/neutrophils interaction. Data suggest that in sepsis manifested in the presence of obesity, the exaggerated inflammatory response and NET formation may be limited due to dysfunctional platelets [123]. The authors further investigated the engagement of different metabolic pathways fueling NETosis in the mentioned model. They showed that eutrophic mice neutrophils release NETs utilizing energy obtained from glycolysis and/or the pentose phosphate pathway under both physiological and inflammatory conditions. However, septic HFD mice neutrophils utilize these pathways only for spontaneous NET formation. Upon secondary ex vivo activation, neutrophils exhibit an “exhausted phenotype”, characterized by the diminished release of NETs despite retaining glycolytic activity and the potential to oxidize fatty acids. These findings highlight that neutrophil NET release is modulated by the individual’s metabolic and inflammatory status [124]. In both studies, only male mice were used.

Given the critical role of NE and MPO in forming NETs, Braster et al. hypothesized that NETs play a role in early adipose tissue inflammation [20]. In a mouse model of HFD-induced obesity, they evaluated the effects of a PAD4 inhibitor Cl-amidine. Ten-week-long treatment affected neither insulin sensitivity or cholesterolemia nor weight gain, organ weight, weight of fat depots, and adipocyte size in eutrophic or obese animals. Moreover, there were no differences in leukocyte, myeloid cell, neutrophil, macrophage, cytotoxic T cells, T helper cell, natural killer cell, and B cell counts or in macrophage phenotype in epidydimal and subcutaneous fat or liver. The authors confirmed the presence of NETs in the adipose tissue of obese mice using confocal microscopy following immuno-fluorescent staining for the neutrophils, cathelicidin, and DNA. It was concluded that in the model of HFD-induced obesity, blocking histone citrullination and subsequent NET release does not affect metabolic variables, leukocyte infiltration, and activation in adipose tissues and liver.

On the other hand, Wang et al. [125] demonstrated that NETosis plays a pathogenetic role in obesity-induced vascular endothelial dysfunction—an early forerunner of atherosclerosis [126,127]. In male mice, obesity induced by HF-HSD administration was associated with higher plasma glucose, insulin, monocyte chemoattractant protein-1 (MCP-1), and vascular cell adhesion molecule 1 (VCAM-1), but not IL-6 concentrations. The presence of NETs in mesenteric arterioles of obese animals was confirmed by higher staining by cathelicidin-related antimicrobial peptide (amidinemidine)—a surrogate marker of NET formation. After 8 and 9 weeks of HF-HSD, daily treatment of obese mice with Cl-amidine (an inhibitor of PAD4) or DNase (degrading NETs), respectively, was initiated. Neither the 2-week-long administration of Cl-amidine nor the 1-week-long treatment with DNase affected adiposity, glucose metabolism, IL-6 or VCAM-1 levels, but they markedly reduced MCP-1 levels and CRAMP immunostaining. Moreover, both interventions restored the acetylcholine-induced endothelium-dependent vasorelaxation response of mesenteric arterioles, which was significantly impaired in obese vs. lean mice. These findings identify NET formation as a driver of obesity-induced endothelial dysfunction, which can be restored by targeting NETosis independently of weight loss or improvement of glycemic status. However, the relationship between NETosis and endothelial dysfunction is, with high probability, bidirectional. NETosis may induce endothelial dysfunction via pro-inflammatory and pro-thrombotic pathways, while vascular impairment, particularly in an inflammatory setting, can trigger NET formation as an immune response to injury [128,129]. Understanding this relationship is important for therapeutic strategies to restore vascular health and prevent associated complications via targeting NET formation or the underlying causes of endothelial dysfunction.

Obesity is an independent risk factor for complications arising from severe influenza infection [130]. Thus, Moorthy et al. studied whether obesity is associated with higher pulmonary NET formation that aggravates the outcome of influenza pneumonia [131]. Male BALB/c mice administered low or HFD for 18 weeks were intratracheally administered either a lethal dose of PR8 virus or phosphate-buffered saline and sacrificed 6 or 10 days later. Mice on HFD had higher body weight and adiposity, while glycemia was similar between the groups. Upon lethal challenge, body weight loss, pulmonary viral load, and lung pathology were similar in both groups. In lung sections, NETs were quantified by triple immunolabelling with DAPI, antibodies against histone H2B, and MPO. The NET score was significantly higher in HFD-administered mice than in controls, with a tendency toward worse scores in obesity-infected animals. Thus, in the murine model of severe influenza pneumonia, adiposity leads to a relatively higher formation of NETs in the lungs.

Asthma with obesity is a specific phenotype, as obesity increases the risk of asthma development and worsens its severity and control, as it is resistant to steroids [132,133]. The obese-combined asthma model was established in male mice by feeding an HFD for 12 weeks and repeated intranasal administration of house dust mites (HDM) [134]. Animals with asthma fed an HFD displayed higher body weight, adiposity, total-, LDL-cholesterol, triacylglycerols, total IgE, HDM-IgE, and lower LDL-C levels. They showed symptoms of airway inflammation such as nose scratching, and choking and airway hyper-responsiveness with increased airway resistance. In bronchial lavage fluid (BALF), several pro-inflammatory cytokines were higher than in eutrophic healthy controls. Lungs were infiltrated by inflammatory cells, and the relative optical density of immunostained NETS (with DAPI, anti-citrullinated histone 3 ab, and anti-MPO ab) was >2-fold higher than in controls. At week 12, treatment with Involucrasin B (IB, a dihydroflavonoid of a plant origin) or dexamethasone administered daily for 1 week by intragastric gavage was initiated. Neither treatment affected body weight or adiposity significantly. Administration of IB, but not dexamethasone, improved lipid profile. Both treatment modalities ameliorated airway resistance, increased lung infiltration of inflammatory cells and mucus secretion, and NET formation. Treatment with IB improved the pro-inflammatory profile, while dexamethasone did not significantly affect BALF’s IL-8 and TNF levels. Both treatment modalities attenuated neutrophilic-type lung inflammation, reflecting that the formation of NETs is related to the secretion of inflammatory cytokines.

Obesity increases the risk of developing pancreatic cancer 2- to 4-fold—one of the most lethal malignancies with a five-year survival rate of <10% [135,136]. Two studies focused on the effects of obesity on pancreatic carcinogenesis and the potential of metformin to rescue the effects of obesity and NETs. The first study established a tumor model by implanting murine Lewis Lung Carcinoma cells in male mice fed either a standard or HFD [137]. Obese tumor-bearing mice had higher body weight and more aggressive tumor growth. They displayed obesity-associated metabolic and pro-inflammatory alteration, such as higher fasting glycemia, insulinemia, insulin resistance, leptin, glutamine, soluble P-selectin levels, transforming growth factor-β (TGF-β1), and high-mobility group box-1 (HMGB1) levels as well as leukocyte, lymphocyte, and neutrophil counts than the standard diet-fed lean mice. Obese tumor-bearing mice had higher levels of NETs in circulation (estimated by analysis of dsDNA and citrullinated histone H3) and tumor tissue (immunohistochemical analysis using anti-PAD4, anti-MPO, anti-citrullinated histone H3, and anti-NE antibodies). Metformin ameliorated all metabolic and pro-inflammatory alterations in tumor-bearing HFD-fed obese mice, but glutamine even increased under metformin treatment. Metformin reduced tumor growth, probably via improving elevated glycemia, insulinemia, soluble P-selectin, TGF-β1, HMGB1, and tumor expression of malignance-associated signaling molecules, such as those promoting proliferation, and aerobic glycolysis-, glutaminolysis-, platelets-, and neutrophils-associated molecules. In ex vivo studies with human neutrophils, metformin mildly ameliorated spontaneous NETformation and counteracted NETosis induced by HMGB1. Moreover, in in vitro experiments, cancer cells promoted NET formation in a cell contact-independent manner in response to HMGB1. In vitro, metformin decreased LLC cell viability and migration and reduced the expression of proliferation and metabolism-related signaling molecules in LLC cells.

Wang et al. employed a genetic model of ductal carcinoma (*Pdx1-Cre*; *LSL-KrasG*^12*D*^^+/−^, KC mice) in female mice and administered a standard or HFD concurrently with orally administered metformin or placebo [138]. After 30 weeks, animals on HFD displayed higher body and pancreas weight; plasma concentrations of insulin, glucose, triglycerides, cholesterol; and pancreas and liver steatosis than controls. Moreover, pancreases of HFD mice were infiltrated by large adipocytes. All these features were significantly ameliorated by metformin administration in obese animals, while in mice fed a standard diet, no significant effect of metformin was observed. In the pancreas, obesity promoted fibrosis, intraepithelial neoplasia (mPanINs) formation, and accumulation of NETs, which were significantly alleviated by metformin. Moreover, administration of DNase I inhibited mPanINs progression and decreased CK-19 positive ductal lesions and collagen deposition, suggesting a tumor-promoting role of NETs in obese mice. In vitro studies showed that NETs promote the proliferation capacity and epithelial to mesenchymal transition in mPanINs cells of obese mice via TLR4-dependent pathways, thereby contributing to disease progression.

#### 4.2.3. Summary

The association between obesity and NETosis is evident, with NETosis acting as both a contributor to and a consequence of the chronic inflammatory state. This interplay may play a critical role in the pathogenesis of obesity-related complications, e.g., obesity alters the balance of immune cell subsets toward a pro-inflammatory phenotype. In neutrophils, this change facilitates NETosis. Expanded adipose tissue secretes chemokines, which recruit neutrophils to the tissue and release pro-inflammatory cytokines, activating neutrophils and promoting NETosis. Thus, adipose tissue in obesity contains higher numbers of neutrophils, which can release NETs and exacerbate inflammation. NETs play a role in the development of obesity-related complications, such as atherosclerosis and thrombosis, and contribute to the increased risk of cardiovascular events in individuals living with obesity by affecting immune and metabolic pathways.

Given the role of NETosis in obesity-related inflammation and metabolic disorders, targeting NETosis is being explored as a potential therapeutic strategy. Approaches include direct ones—either aimed to inhibit NET formation via targeting PAD4, a key calcium-dependent enzyme in NET formation; or those boosting NET degradation via administration of DNase, degrading ecDNA; or indirect ones targeting the pro-inflammatory environment. Experimental studies indicate that the antidiabetic drug metformin exerts an anti-NETotic effect by influencing cellular energy metabolism and inflammatory pathways, thus suppressing NETosis through direct and indirect pathways.

## 5. Neutrophil Extracellular Traps in Hypertension

Hypertension is the most important but treatable risk factor for CVD and death worldwide. Its increasing prevalence is attributed to population growth, aging, and behavioral factors, such as unhealthy diet, harmful alcohol consumption, physical inactivity, and excess weight, but also to the increasing prevalence of patients with immune-mediated disease [139,140]. Immune response and inflammation increase the risk of hypertension manifestation and associated mortality [141]. In the large cross-sectional UK Biobank Study (>384 thousand participants), elevated BP was associated with higher counts of neutrophils, monocytes, and lymphocytes, but the associations of neutrophil numbers with SBP, DBP, and pulse pressure were the strongest compared with the other white blood cells analyzed [142]. In Japanese females with neutrophil counts within the reference range at baseline, hypertension incidence over 40 years showed a significant association with higher neutrophil count [143]. The neutrophil-to-lymphocyte ratio increases with rising BP and correlates with the risk of developing hypertension [144]. Moreover, plasma levels of pro-inflammatory cytokines and chemokines also correlate directly with BP [145,146,147,148,149]. Emerging evidence suggests that NETs contribute to the development and progression of hypertension through their pro-inflammatory, pro-thrombotic, and vascular-damaging properties. Elevated BP is an essential component of MS, defined as the presence of central obesity and any two out of the other four CVD risk markers, e.g., impaired glucose metabolism, hypertension, and atherogenic dyslipidemia (elevated triacylglycerols or low HDL-C levels) [150]. Albeit obesity is a state of low-grade inflammation associated with higher NET formation, data on NETosis in obese patients with elevated BP, or hypertensive patients per se, are scarce.

### 5.1. Human Studies on Hypertension and NET Formation

Neutrophil elastase was significantly higher in patients with hypertension than the control subjects, regardless of age and neutrophil counts [151]. In a small cross-sectional study (n = 10, each group), obese pre-hypertensive females displayed higher serum concentrations of NE than their obese normotensive peers, and both groups presented with higher levels than non-obese normotensive females. In the pre-hypertensive obese group, NE concentrations correlated directly with BMI, waist circumference, systolic and diastolic BP, total cholesterol, triacylglycerols, and C-reactive protein (CRP), and inversely with HDL-C and markers of pulmonary function [152]. The authors suggest that in patients living with obesity, elevated serum NE levels reflect the presence of low-grade inflammation and might contribute to the development of prehypertension and lung function impairment. However, circulating NE levels are not an unequivocal indicator of NETosis. NE (E.C. 3.4.21.37) has a multitude of potential substrates, such as almost all components of the extracellular matrix, clotting factors, complement, immunoglobulins, and cytokines [153]. Intracellular NE is required for the effective killing of phagocytosed bacteria. Extracellular NE may either act as a negative modulator of the inflammatory response due to its ability to degrade cytokines, cell surface receptors, and complement components or induce tissue damage under unregulated inflammation [154]. Since this unopposed action of NE might be detrimental, the organism possesses several endogenous inhibitors of NE, such as α_1_-antitrypsin, α_2_-macroglobulin, elafin, or secretory leukoproteinase inhibitor [155]. Patients with pulmonary hypertension (n = 249) display higher circulating levels of NE and lower levels of elafin than healthy subjects (n = 106) [156].

Naïve patients with essential hypertension (EHT, n = 55) displayed higher levels of NETs in plasma (determined as MPO/DNA complexes and citH3 by ELISA methods) compared with their normotensive peers (n = 26), and plasma samples from untreated EH patients primed neutrophils isolated from healthy individuals to form NETs in an ROS and histone citrullination-dependent manner [157]. NET release in plasma significantly correlated with plasma thrombin–antithrombin activity, suggesting the thrombogenic potential of NETs in EH. In vitro, angiotensin II (AT II) enhanced the NETosis via an ROS/autophagy-dependent manner, and PAD4 histone citrullination was associated with Ang II-induced NETosis. Finally, they showed that in EH, Ang II links neutrophils and NETosis with thromboinflammatory collagen production in activated endothelial cells, eventually promoting vascular injury and interstitial renal fibrosis.

Similarly, in neutrophils of healthy donors, ex vivo NET formation was abolished after pre-treatment with the AT II type 1 receptor blocker (losartan) [158]. The authors also reported a positive correlation between BP and ex vivo-induced NETosis in neutrophils isolated from patients with coronary heart disease. These findings indicate that NETosis is influenced by BP and AT II, providing a link between inflammation and the pathogenesis of cardiovascular disease.

Li et al. suggested a link between increased coagulation activity and NET levels in EHT [159]. Patients with moderate to severe EHT showed higher plasma NET concentrations than those presenting with mild forms of disease or normotensive controls. Pro-coagulant effects of NETs (decreasing clotting time and increasing potency to generate thrombin and fibrin) were effectively attenuated by co-incubation with DNase I. Moreover, NETs from patients with moderate to severe EHT exerted a strong cytotoxic effect on endothelial cells.

Higher circulating NET levels compared with healthy controls were also documented in patients with pulmonary arterial hypertension [160]. It was concluded that NETs trigger inflammatory activation of pulmonary endothelial cells and promote endothelial angiogenesis via MPO/H_2_O_2_/nuclear factor κB/TLR4-dependent signaling. Their pathogenetic role in inflammatory angiogenesis relevant to arterial hypertension was implicated.

### 5.2. NETosis in Rodent Models of Hypertension

To confirm the hypothesis that NETs promote changes in vasculature leading to hypertension, Fang et al. compared the accumulation of NETs in mesenteric arteries of male spontaneous hypertensive (SHR) and Wistar-Kyoto rats [161]. Accumulation of NETs (detected as citH3) in the vessel wall of SHR was significantly higher than in their normotensive counterparts. Repeated injections of NET inducer PMA over 3 months into the tail vein of male BALB/c mice induced hypertension and was associated with higher expression of NET markers (MPO, citH3), suggesting that NET formation relates to increased BP. In vitro studies showed that NETs promote the G1/S transition and facilitate vascular smooth muscle cell (VSMC) proliferation via the PI3K/Akt/CDKN1b-signaling axis. The rapid proliferation of VSMCs induced by NETs seemed TK1-dependent and transferred via exosomes. Thus, NETs-driven phenotypic alterations of VSMCs may play a pathogenetic role in developing arterial hypertension.

Portal hypertension induced by the partial ligation of suprahepatic inferior vena cava in C57BL/6J mice resulted in the accumulation of neutrophils, platelets, and NETs in the liver sinusoids, partially contributing to the increase in portal pressure by promoting sinusoid microthrombi and fibrosis [162].

### 5.3. Summary and Perspectives

Due to the complex interplay between immune responses, vascular biology, and the pathophysiology of hypertension, studying hypertension-associated NETosis presents several challenging questions. However, only a few studies focus on this issue, suggesting that higher neutrophil counts (within their reference range) are associated with higher BP in the general population. Data on the association of circulating NE with BP are equivocal. Albeit data indicate that patients with EHT display higher levels of circulating NETs than normotensive subjects, it remains unclear whether this is a causal association with elevated BP or potentially just reflects the presence of other cardiovascular risk factors or markers. Studies in adolescents with EHT, free from comorbidities associated with aging, and longitudinal studies in the SHR or DOCA-salt model could elucidate whether increased NETosis is a forerunner or a consequence of hypertension. The same questions apply to obesity-associated elevated BP. If the formation of NETs in hypertension results from mechanical stretch, AT II signaling, and increased ROS production, the subsequent activation of endothelial cells could drive vascular dysfunction and remodeling, making NET-related pathways potential therapeutic targets for hypertension. In this case, pharmacological blocking of NETosis or degradation of NETs via administration of DNase should control elevated BP. The challenging question remains as to whether achievement of satisfactory BP control itself goes hand in hand with the normalization of enhanced NETosis or whether NETs would be affected only by selective treatment modalities, such as renin–angiotensin–aldosterone system inhibitors and AT II receptor 1 blockers (as neutrophils express AT receptors) or calcium channel blockers (since some types on NETosis require external calcium flux) or whether administration of metformin to obese patients or in obesity-associated HT model would mitigate NETosis.

## 6. Dyslipidemia and NET Formation

Atherogenic dyslipidemia creates a vicious cycle of lipid abnormalities, inflammation, and vascular damage, ultimately leading to severe cardiovascular and systemic complications. Dyslipidemia triggers vascular remodeling; endothelial dysfunction; chronic inflammation; atherogenesis; atherosclerosis; chronic inflammation; and, among others, the manifestation of metabolic syndrome and worsening its other components, or that of metabolic dysfunction-associated steatotic liver disease (MASLD) [163,164].

Atherogenic dyslipidemia and NETosis are interlinked in a vicious cycle of inflammation and vascular damage. Dyslipidemia promotes NETosis, which in turn exacerbates atherosclerosis and its complications. Understanding this relationship opens avenues for innovative therapies targeting both lipid abnormalities and neutrophil-driven inflammation.

### 6.1. Human Studies on NETs in Steatohepatitis

Neutrophil infiltration is critical in mediating the transition from steatosis to nonalcoholic steatohepatitis (NASH) [165]. Thus, Du et al. hypothesized that NETosis plays a pathogenetic role in NASH. Compared to healthy controls, patients exhibited higher plasma levels of NETosis markers, including cfDNA, MPO–DNA, citH3–DNA, and NE–DNA complexes. Plasma from NASH patients induced more pronounced NETosis in freshly isolated neutrophils from healthy donors compared to plasma from controls, and the frequency of NETotic neutrophils was significantly higher in the NASH milieu. Patients also had elevated plasma concentrations of IL-1 and TNF-α, which directly correlated with circulating NET markers. Supplementing NASH plasma with pro-inflammatory cytokines further enhanced NET formation while adding anti-IL-6 and anti-TNF-α antibodies to patient plasma reduced NETosis. Levels of thrombin–antithrombin complexes and fibrinogen showed a direct correlation with NET markers, whereas coagulation time displayed an inverse relationship. Finally, NETs converted HUVECs into pro-coagulant and pro-inflammatory phenotypes, which could be partially mitigated by DNase I treatment. DNases degrade ecDNA into smaller fragments to inhibit immune system activation and consequently reduce the risk of sterile inflammation [166]. These findings suggest that in NASH patients, NETosis is driven by inflammatory factors and contributes to the hypercoagulable state that is characteristic of the disease.

Arelaki et al. analyzed liver biopsy specimens from healthy controls and patients with non-alcoholic fatty liver disease (NAFLD), including specimens with NASH [167]. NETs were identified as extracellular structures based on co-localization of NE and citH3 using confocal microscopy. IL-1β and IL-17A were also assessed. NETs were observed in 16/17 of the NASH biopsy specimens but were absent in all other NAFLD and control samples. The presence of NETs correlated with steatosis, ballooning degeneration, lobular and portal inflammation, NAS score (the sum of scores for steatosis, lobular inflammation, and ballooning), stage, and the diagnosis of NASH. IL-1β and IL-17A were co-localized with NETs. Platelet aggregates were significantly larger in NASH specimens compared to controls. It has been suggested that NETs play a pathogenetic role in NASH and co-localize with pro-inflammatory cytokines.

### 6.2. Rodent Studies on NETs in Steatohepatitis

In the mouse model of NASH induced by methionine and choline-deficient diet (MCD), the recruitment of neutrophils is an early inflammatory event, followed by the formation of NETs [168]. It has been shown that the metabolites in linoleic acid could prime neutrophils to undergo NETosis through oxidative burst. Inhibition of NETosis by administration of DNase I or silybin (a hepatoprotective agent alleviating NASH by reducing liver lipid accumulation and fibrosis [169]) ameliorated inflammation and blocked NET formation. These data underline the metabolism-immune causal link in NASH progression and demonstrate the importance of inhibitors of NETs in treating NETosis-related diseases.

In the same dietary model, NET accumulation in the livers of mice was effectively ameliorated by daily injection of Tanshinone IIA—one of the main active components in *Salvia miltiorrhiza*—used in the treatment of cardiovascular diseases due to its antioxidant, anti-inflammatory, and antiatherosclerotic effects [170].

Albeit HFD or HF-HSD diets induce obesity-associated liver steatosis in mice and rats [171,172,173,174], to our knowledge, NETosis has not been explored in these models. Additionally, female offspring of dams in whom obesity was induced preconceptionally by administration of HFD present with liver steatosis and nonalcoholic fatty liver disease, whereas males develop liver fibrosis and manifest nonalcoholic steatohepatitis, but data on NET formation are lacking [175,176,177].

### 6.3. Summary and Perspectives

With the rising prevalence of obesity and obesity-associated metabolic syndrome, metabolic dysfunction-associated steatotic liver disease (MASLD) is also on the rise. HFD and cafeteria diet-induced obesity models, probably mimicking liver steatosis pathogenesis in experimental rodents better than deficiency diet, are more suitable for studying the role of NETosis. It remains unclear whether NETs occur in circulation and the liver concurrently, whether gut dysbiosis affects NETosis and contributes to liver steatosis, whether NETs play a role in the progression of steatosis to fibrosis, and whether they affect Kupffer cell function. It also remains unclear as to whether there is an interplay between liver NETosis and insulin resistance.

## 7. Knockout Animal Models in the Obesity and Metabolic Syndrome Research

### 7.1. PAD4 Deficiency

In an HFD-induced obesity model in Wild-type and neutrophil-selective PAD4 knockout (Ne-PAD4^(−/−)^) mice, the role of peptidylarginine deiminase 4 (PAD4) activity in relation to NET formation and cardiac health was studied [178]. HFD intake induced significant changes in neutrophil priming, favoring NET release, including the early stages of speck formation and histone citrullination of apoptosis-associated speck-like protein containing a CARD. Ne-PAD4^(−/−)^ mice, which do not form NETs, gained, throughout a 10-week experiment, less weight than their wild-type counterparts. While obesity progression led in wild-type mice to cardiac remodeling and diastolic dysfunction, these abnormalities were not manifested in Ne-PAD4^(−/−)^ mice. However, HFD did not affect NET levels or thrombus formation in the inferior vena cava stenosis model. Thus, in the context of chronic inflammation, outcomes of obesity are only partially driven by neutrophil PAD4 activity and NET release (Figure 2A).

To study the role of NETs formation in hypertension, Krishnan et al. used male PAD4-deficient (PAD4^(−/−)^) and C57Bl/6 mice, administered AT II via subcutaneous minipump for 28 days [179]. The rise in BP in response to AT II was lower in PAD4-deficient than wild-type animals. In response to saline injection, PAD4-deficient mice showed a trend for improvement in renal sodium and urinary volume excretion. Harvested mesenteric arteries from PAD4-deficient mice showed improved vascular relaxation in response to acetylcholine. At the same time, in the aortas, infiltration with neutrophils, dendritic, and cytotoxic CD8+ T cells was reduced compared with C57Bl/6 mice. To investigate the potential impact of endothelial cell (EC) stretch on NETosis, primary neutrophils were cocultured with or without ECs. ECs enhanced neutrophil survival and promoted NETosis. When exposed to hypertensive uniaxial stretch, NETosis significantly increased, and coculturing neutrophils with ECs under hypertensive stretch further amplified this effect. Moreover, neutrophils subjected to hypertensive stretch displayed higher histone citrullination. Hypertensive stretch also enhanced suicidal NETosis, potentially intensifying the inflammatory response in ECs. These results indicate that NETosis plays a role in hypertension, aortic inflammation, and related EC dysfunction, with EC stretch serving as a direct inducer of NETosis.

### 7.2. MPO Deficiency and Inhibition

MPO is a member of the peroxidases subfamily and is most abundant in neutrophils [180]. MPO is stored in the cytoplasmatic membrane-bound azurophilic granules, secreted in response to various stimuli (mostly pathological agents) into the extracellular space by degranulation or exocytosis [153]. MPO catalyzes the oxidation of chloride and other halide ions in the presence of H_2_O_2_, generating hypochlorous acid and other highly reactive products that confer efficient antimicrobial action [181]. The MPO is released from the neutrophils during the NET formation and is thus considered one of the markers of NETosis. To our knowledge, plausible data on how selective knockout of MPO or MPO-deficiency affects NET formation in different mice models of obesity and associated health conditions are not available. Here, we summarize the usage of MPO-deficient mice in the components of metabolic syndrome. 

Administration of oral MPO inhibitor (2-thioxanthine AZM198) to obese hypertensive (induced by an HFD and ATII infusion) C57BL/6J mice reduced elevated MPO levels to normal values, ameliorated body weight gain, fat accumulation, reduced visceral adipose tissue inflammation, and the severity of nonalcoholic steatohepatitis [182]. Furthermore, MPO-knock-out mice subjected to 5/6 nephrectomy displayed lower renal damage (albuminuria, glomerular injury, and renal inflammation) and lower plasma MPO levels than WT mice. At the same time, BP did not differ significantly between the groups [183]. Thus, in animal models, MPO deficiency or its inhibition attenuates cardiometabolic risk factors or markers. Further studies are needed to reveal whether amelioration of NETosis plays a pathogenetic role in these beneficial effects. 

Despite these promising data, further studies are required to understand whether and how MPO inhibition could be used as a novel potential treatment of metabolic syndrome (Figure 2B).

### 7.3. Neutrophil Elastase Deficiency

NE is a neutrophil-specific serine protease secreted from the primary granules while regulating NET formation [184]. Several studies documented that NE is crucial in obesogenic-diet-induced inflammation [99,185,186]. Here, we summarize studies focused on the effects of deleting NE on components of metabolic syndrome.

Talukdar et al. studied NE-deficient mice who consumed HFD for 12 weeks. Although body weight gain did not differ between the NE-deficient and wild-type mice, NE deletion or its inhibition using an inhibitor improved glucose metabolism in HFD-fed mice via increasing insulin sensitivity. Under hyperinsulinemic–euglycemic clamp, NE-deficient mice on HFD displayed increased glucose infusion rate, decreased hepatic glucose production, and increased suppression of hepatic glucose. Additionally, the FFA concentrations decreased after fasting and clamping procedures, with increased suppression of lipolysis. The authors concluded that NE deletion improves the inflammation response in the adipose tissue and liver that occurs in mice under HFD consumption, as well as insulin sensitivity and signaling primarily via the liver [99]. On the other hand, Mansuy-Aubert et al. showed that NE deletion rescues mice from body weight gain after 10 weeks of HFD consumption. Lower body weight gain was mirrored by smaller adipocyte size in epididymal fat tissue and lower total epididymal and inguinal fat tissue. Additionally, NE-deficient mice on HFD did not develop liver steatosis (Figure 2B); they showed lower amounts of triacylglycerols in the muscles, liver, and serum. Notably, NE-deficient HFD-fed mice showed higher energy expenditure and rectal temperature than their wild-type peers. The authors confirmed that NE deletion improves insulin sensitivity and adipose tissue inflammation under the HFD [186]. However, another study using NE deletion in HFD mice suggested that NE signaling in vascular endothelial cells increases para-endothelial permeability to immune cells, thus accelerating inflammatory tissue damage, as observed in obese animal models [187].

## 8. Discussion and Perspectives

Studies on NETosis indicate that obesity is associated with heightened NET formation at baseline (indicating spontaneous NETosis) and ex vivo, in response to external stimuli. The association between obesity and NETosis is bidirectional, with NETs acting both as inducers/contributors to and a consequence of the chronic inflammatory state. NETs are considered promising candidates to link the chronic inflammatory state of obesity with the development of metabolic and cardiovascular complications. However, more experimental and clinical research is needed on the pathomechanisms of this relationship to reveal whether circulating NETs are risk factors or markers.

Some questions regarding basic physiology remain to be elucidated. First, we lack data on the intraindividual variability of plasma NET levels, their dynamics during the lifespan (physiological aging), and potential sex differences. Rodent models are a method of choice to study long-term effects. Still, sex differences should be verified in human studies since neutrophils-to-lymphocyte ratios are opposite in humans and rodents. Second, potential NET structure and composition variations between lean subjects and those living with obesity need to be elucidated as these factors might affect NET’s functional role. Third, firm evidence documenting that levels of circulating NETs reflect their accumulation in VAT and other tissues is needed. This knowledge is important for deciding whether plasma levels of NETs could be used as diagnostic or prognostic criteria. It should be documented whether increased spontaneous NETosis is primarily a marker (early or late) of VAT inflammation or, more complexly, reflects an increased cardiometabolic risk. The pathophysiological role of NETosis could be verified via several approaches, such as pharmacological interference with neutrophils’ ability to form NETs. However, non-selective inhibitors may lead to off-target effects. Administration of DNase would lead to the degradation of existing NETs into shorter fragments with lower immunogenicity. A shorter size of NET remnants would also facilitate their clearance by macrophages or the kidney. However, post-translational modification of mitochondrial NETs, such as histone citrullination or cross-linking of mitochondrial DNA with proteins, render mitochondrial NETs resistant to degradation by DNase. The question arises as to whether NETosis could be mitigated by administering anti-platelet drugs, thus diminishing neutrophil–platelet interactions. Clarifying these issues could elucidate whether NETs cause or are a consequence of obesity-associated inflammation and risk factors or are just markers of increased cardiometabolic risk.

In this context, studies on NETosis employing animal models present a valuable perspective to uncover the complexity of NETosis by identifying its molecular pathways and outlining potential new therapeutic targets and approaches. Existing and new knock-out models offer an approach to dissecting the role of a particular pathway, receptor, or their combination.

## Figures and Tables

**Figure 1 nutrients-17-00241-f001:**
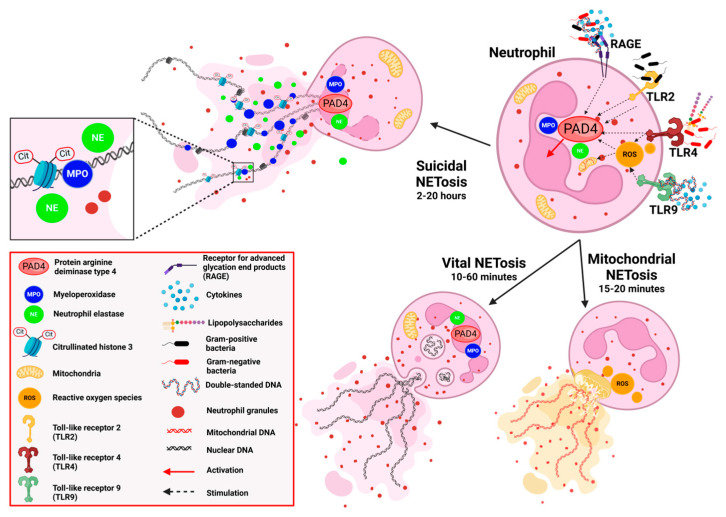
The main forms of NETosis: suicidal, vital, and mitochondrial. Created with BioRender.com.

**Figure 2 nutrients-17-00241-f002:**
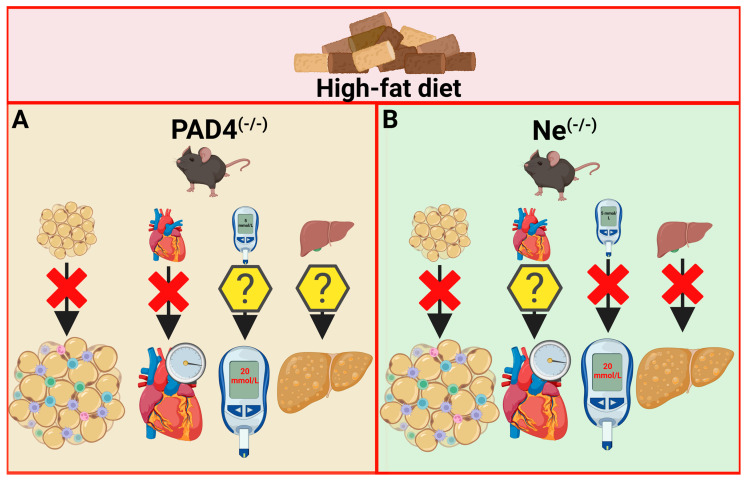
NETs formation in response to obesogenic diet in knock-out mouse models. (**A**) The Protein Peptidyl deiminase type 4 (PAD^(−/−)^) deficiency shows a protective role in the development of obesity and hypertension (X). However, the protection against the development of insulin resistance and steatohepatitis remains unknown (?). (**B**) Deleting the neutrophil elastase (Ne^(−/−)^) in mouse lines protects animals from developing obesity, insulin resistance, and steatohepatitis (X), while the effect of NE deficiency on hypertension needs to be elucidated (?). X, blocking the action, e.g., a protective effect; ?, effects unknown/not reported. Created with Biorender.com.

## Data Availability

No new data were created or analyzed in this review. Data sharing is not applicable to this article.

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
