# Peer review of "Pathophysiological Role of Neutrophil Extracellular Traps in Diet-Induced Obesity and Metabolic Syndrome in Animal Models"

_nutrients, 2025, doi:10.3390/nu17020241_

Round 1
Reviewer 1 Report
Comments and Suggestions for Authors
Following a comprehensive introduction to obesity, the concept of NETosis, and NET formation, the review extensively and, at times, exhaustively discusses the role of neutrophils in general, their phenotype, and, most notably, NETosis in obesity and its major comorbidities. The review examines studies conducted in mice, and although these are understandably more limited, it also includes studies in humans. After each section, the main conclusions of the studies are clearly summarized for the reader. Overall, this is an extensive and well-conducted review on the role of NETosis in the underlying inflammatory mechanisms of obesity, offering a novel and conceptually important perspective. This is not only due to the role of neutrophils but also the mechanism of NETosis addressed.
However, the text contains some important conceptual errors. For example, on page 2, it states that IL-1 is anti-inflammatory and reduces oxidative stress, when it is clearly a pro-inflammatory cytokine. Additionally, I miss the mention of the intracellular microbicidal capacity of neutrophils, both oxygen-dependent and oxygen-independent, in several paragraphs where their function is discussed.
In addition, I suggest a final paragraph, not only outlining future perspectives but also providing a general conclusion, particularly the authors' opinion regarding the physiological relevance of the topic discussed.
Furthermore, the text should be corrected for language and writing errors in English. For instance, in line 278, the abbreviation "BP" for Blood Pressure should appear earlier in the paragraph. Additionally, the beginning of section 5.3 lacks clarity and coherence. I recommend a thorough revision to address these issues.
I hope these constructive comments will help the authors improve their good manuscript.
Author Response
Thank you for allowing us to submit a revised draft of our manuscript. We appreciate the time and effort that the editors and the reviewers dedicated to providing feedback on our manuscript, and we are grateful for the insightful and valuable comments on our paper.
We have incorporated the suggestions made by the reviewers. Revisions in the text are shown in red for additions or corrections. Please find below our point-by-point response to each of the reviewers' comments. We have tried to follow all of them.
We hope that the revisions in the manuscript and our accompanying responses will be sufficient to make our manuscript suitable for publication in the Nutrients.
Reviewer 1:
Following a comprehensive introduction to obesity, the concept of NETosis, and NET formation, the review extensively and, at times, exhaustively discusses the role of neutrophils in general, their phenotype, and, most notably, NETosis in obesity and its major comorbidities. The review examines studies conducted in mice, and although these are understandably more limited, it also includes studies in humans. After each section, the main conclusions of the studies are clearly summarized for the reader. Overall, this is an extensive and well-conducted review on the role of NETosis in the underlying inflammatory mechanisms of obesity, offering a novel and conceptually important perspective. This is not only due to the role of neutrophils but also the mechanism of NETosis addressed.
Response: We appreciate the reviewer's time spent and effort in pointing out the weaknesses of our paper and her/his constructive attitude in suggesting improvements. We tried to do our best to implement the comments and suggestions.
However, the text contains some important conceptual errors. For example, on page 2, it states that IL-1 is anti-inflammatory and reduces oxidative stress when it is a pro-inflammatory cytokine.
Response: Many thanks for the remark. We apologize for the misprint; we meant IL-10, which was corrected in the revised paper (section 1.1., line 72).
Additionally, I miss the mention of the intracellular microbicidal capacity of neutrophils, both oxygen-dependent and oxygen-independent, in several paragraphs where their function is discussed.
Response: We would like to thank the reviewer for her/his guidance. We have added a short paragraph regarding neutrophils' oxygen-dependent and oxygen-independent microbicidal capacity (section 1.2., lines 82-96).
In addition, I suggest a final paragraph, not only outlining future perspectives but also providing a general conclusion, particularly the authors' opinion regarding the physiological relevance of the topic discussed.
Response: According to the reviewer's suggestion, the discussion and perspectives part of the manuscript was rephrased (section 8, lines 925-945).
Furthermore, the text should be corrected for language and writing errors in English. For instance, in line 278, the abbreviation "BP" for Blood Pressure should appear earlier in the paragraph.
Response: We apologize for overlooking this issue. The abbreviation (BP) is given after first mentioning the blood pressure (section 4.1., line 383).
Additionally, the beginning of section 5.3 lacks clarity and coherence.
Response: We are thankful that the reviewer pointed out this mistake. We forgot to erase the part of the sentence when finalizing the manuscript. We have corrected it in the revised paper (section 5.3., lines 732-749).
I recommend a thorough revision to address these issues. I hope these constructive comments will help the authors improve their good manuscript.
Reviewer 2 Report
Comments and Suggestions for Authors
The manuscript address an area regarding the role of NETs in obesity and associated disorders. The authors review NET formation and its association with diet-induced obesity, inflammation, and metabolic syndrome, in both human and animal studies. The paper is well-structured and describes the relevance of NETosis in the progression of obesity-related complications such as hypertension and liver steatosis.
There are areas that need improvement:
The abstract sets the stage but a summary of the specific pathophysiological mechanisms could be cited in key findings or emerging and trends from animal models.
The introduction is good, but there is redundancy in the first two paragraphs.
Lines 48: The discussion of "metabolically healthy obesity" is ok but could be moved where inflammation is address.
The section 2: is detailed, but some subsections are lengthy. Consider condensing descriptions of NETosis types to focus on their relevance to obesity. Here, a diagram of an ilustrative figure summarizing NETosis mechanisms would improve for readers.
The term "morbid obesity" is used in the manuscript. It is now considered outdated and stigmatizing when describing human subjects. Current guidelines recommend using more appropriate such as "severe obesity" or "individuals with severe obesity"
Lines 307: The authors describe NET concentrations post-bariatric surgery. It would be useful to briefly discuss the implications of NET persistence in a long term period (also in line 328 -334)
Lines 328: The finding of reduced NET formation after weight loss is significant but underexplored. Consider adding a sentence on the broader implications for immune restoration.
Lines 344: The discussion on diet models is relevant, but this section could be condensed. Authors need to focus more on the physiological relevance of each model to human obesity.
Lines 416: The role of NETs in endothelial dysfunction is good. But the authors could clear if NETosis is a cause or consequence of vascular impairment.
The section of hipertensio: The role of neutrophil elastase is discussed without mentioning potential inhibitors. Needs the inclusuin of therapeutic perspectives. And need carefull because do not provide sufficient evidence to support the role of NETs and increased blood pressure.
NETs and Liver Steatosis: The section os steatosis discuss the transition from steatosis to NASH. Authors mention DNase I and silybin as interventions but do not elaborate.
The conclusion summarize key findings but need to emphasize gaps in the literature and point out criticism in the area. Also, it remains unclear whether NET is a cause or a consequence of obesity-induced inflammation.
The authors describe NETosis contributing to systemic inflammation but omit the interplay with other immune cells. Needs improvement.
Line 369 – need to be carefull when citing cafeteria diets – Authors fail to address the variability and reproducibility associated with cafeteria diets. Why cafeteria diets better mimic human obesogenic condition? Need clarification
Author Response
The manuscript address an area regarding the role of NETs in obesity and associated disorders. The authors review NET formation and its association with diet-induced obesity, inflammation, and metabolic syndrome, in both human and animal studies. The paper is well-structured and describes the relevance of NETosis in the progression of obesity-related complications such as hypertension and liver steatosis.
There are areas that need improvement:
Response: We appreciate the time spent and the reviewer's effort to point out our paper's weaknesses and her/his constructive constructive suggestions for improvement. We tried to do our best to implement her/his comments and suggestions.
The abstract sets the stage but a summary of the specific pathophysiological mechanisms could be cited in key findings or emerging and trends from animal models.
Response: We appreciate the reviewer's suggestions for modifying the abstract. Since this is an abstract of a review article, not a research paper, we believe that a summary of the topics covered is suitable and consistent with abstracts of reviews.
The introduction is good, but there is redundancy in the first two paragraphs.
Response: As suggested, we tried to condense the message in the first two paragraphs, avoiding redundancy (section 1, lines 28-58). We have also added information regarding geographical variations in obesity prevalence, as suggested by reviewer 3.
Lines 48: The discussion of "metabolically healthy obesity" is ok but could be moved where inflammation is address.
Response: Following the reviewer's suggestion, the paragraph on "metabolically healthy obesity" has been moved to section 1.1. Obesity-associated low-grade inflammation (lines 71-80).
The section 2: is detailed, but some subsections are lengthy. Consider condensing descriptions of NETosis types to focus on their relevance to obesity. Here, a diagram of an ilustrative figure summarizing NETosis mechanisms would improve for readers.
Response: Following the reviewer's suggestion, we condensed this section, mainly in parts 2.1. and 2.2. In addition, to follow the reviewer's suggestion, a scheme was prepared regarding different NETosis mechanisms (section 2.2., Figure 1).
The term "morbid obesity" is used in the manuscript. It is now considered outdated and stigmatizing when describing human subjects. Current guidelines recommend using more appropriate such as "severe obesity" or "individuals with severe obesity".
Response: We are thankful to the reviewer for this remark. Throughout the revised paper, we replaced the expression “morbid” with the suggested “severe.” (lines 282, 295, 341, 360, 379, 400).
Lines 307: The authors describe NET concentrations post-bariatric surgery. It would be useful to briefly discuss the implications of NET persistence in a long term period (also in line 328 -334).
Response: As suggested by the reviewer, we elaborated on the potential consequences of long-term NET persistence after bariatric surgery (section 4.1.1, lines 417-432).
Lines 328: The finding of reduced NET formation after weight loss is significant but underexplored. Consider adding a sentence on the broader implications for immune restoration.
Response: To follow the reviewer's suggestion, we discuss the impact of the decrease in NET formation after bariatric surgery in section 4.1.1. (section 4.1.1, lines 417-432).
Lines 344: The discussion on diet models is relevant, but this section could be condensed. Authors need to focus more on the physiological relevance of each model to human obesity.
Response: As suggested, we elaborated on section 4.2.1. to explain the difference in the outcomes of the two diets concerning modeling human diet-induced obesity (lines 435-470).
Lines 416: The role of NETs in endothelial dysfunction is good. But the authors could clear if NETosis is a cause or consequence of vascular impairment.
Response: As suggested by the reviewer, we added sentences discussing this issue in section 4.2.2. (lines 534-541).
The section of hypertension: The role of neutrophil elastase is discussed without mentioning potential inhibitors. Needs the inclusion of therapeutic perspectives. And need carefull because do not provide sufficient evidence to support the role of NETs and increased blood pressure.
Response: As suggested, in the revised paper, we discuss the associations of neutrophile counts with hypertension in more detail. We address the issue of NE inhibitors and highlight that the evidence supporting the role of NETosis in hypertension is insufficient. Additionally, we have also expanded the discussion on potential therapeutic targets (section 5.1., lines 673-684, 696-701; section 5.3., lines 733-749).
NETs and Liver Steatosis: The section of steatosis discuss the transition from steatosis to NASH. Authors mention DNase I and silybin as interventions but do not elaborate.
Response: As suggested by the reviewer, we added a brief explanation of the mechanisms of action of DNase and silybin (section 6.1., lines 784-786; and 6.2., lines 804-805).
The conclusion summarize key findings but need to emphasize gaps in the literature and point out criticism in the area. Also, it remains unclear whether NET is a cause or a consequence of obesity-induced inflammation.
Response: According to the suggestion of the reviewer, the discussion and perspectives part of the manuscript was rephrased (section 8).
The authors describe NETosis contributing to systemic inflammation but omit the interplay with other immune cells. Needs improvement.
Response: We are thankful to the reviewer for pointing out this aspect. As suggested, we briefly discuss the issue in section 2.4. (lines 259-275).
Line 369 – need to be carefull when citing cafeteria diets – Authors fail to address the variability and reproducibility associated with cafeteria diets. Why cafeteria diets better mimic human obesogenic condition? Need clarification.
Response: As suggested by the reviewer, in the revised paper, we focus on explaining how the cafeteria diet mirrors a human Western-type diet in terms of its composition and effects. We agree with the reviewer that until recently, cafeteria diets varied in composition substantially. However, we underline that a recent proposition of cafeteria diet protocol by Lalanza and Soeren (2021) allows for minimal variability and acceptable reproducibility (section 4.2.1., lines 442-447, 449-457, 462-770).
Lalanza JF, Snoeren EMS. The cafeteria diet: A standardized protocol and its effects on behavior. Neurosci Biobehav Rev. 2021; 122:92-119. doi: 10.1016/j.neubiorev.2020.11.003.
Reviewer 3 Report
Comments and Suggestions for Authors
The manuscript by Feješ et al. provides a comprehensive review of the role of neutrophil extracellular traps (NETs) in the context of diet-induced obesity and metabolic syndrome. The authors have successfully compiled a wealth of information from both human and animal studies, highlighting the implications of NETosis in obesity-associated conditions such as hypertension and liver steatosis. The manuscript is well-structured, and the literature review is thorough, making it a valuable contribution to the field. Here are some comments.
- On the whole, the author lacks the necessary diagrams to help readers understand the content of the review more clearly.
Author Response
The manuscript by Feješ et al. provides a comprehensive review of the role of neutrophil extracellular traps (NETs) in the context of diet-induced obesity and metabolic syndrome. The authors have successfully compiled a wealth of information from both human and animal studies, highlighting the implications of NETosis in obesity-associated conditions such as hypertension and liver steatosis. The manuscript is well-structured, and the literature review is thorough, making it a valuable contribution to the field. Here are some comments.
Response: We sincerely appreciate the time and effort the reviewer has dedicated to carefully reviewing our paper and highlighting its areas for improvement. We value her/his constructive feedback and thoughtful suggestions, and we have made every effort to incorporate her/his comments and recommendations to the best of our ability.
The authors mention the global prevalence of obesity and its economic burden. It would be beneficial to include a brief discussion on the geographical variations in obesity prevalence and how this might influence the global health strategies.
Response: As suggested by the reviewer, in the revised paper, we briefly mention geographical variations in the prevalence of obesity and potential global health strategies (section: 1. Introduction, lines 36-50).
A graphical abstract or a figure summarizing the key points of NETosis in obesity would greatly enhance the reader's understanding and engagement with the manuscript.
Response: Following the reviewer's suggestion, a graphical abstract was added as Figure 2 (line 913, under section 7.3., legend: 916-925).
Section 2.2.3: The discussion on mitochondrial NETosis is intriguing. Could the authors elaborate on the potential differences in NET formation between obese and non-obese individuals, particularly in the context of mitochondrial function?
Response: Based on the reviewer's comments, we discuss the potential role of mitochondrial NETosis in obesity-associated sterile inflammation in section 2.2.3. (lines 219-231).
Section 3.1: The human studies cited here are compelling. However, the authors might consider discussing any potential limitations or biases in these studies, such as sample size or population demographics.
Response: As suggested by the reviewer, we discuss the outcomes, pointing out the small number of studies run on a small number of probands – particularly females (section 3.1 and 3.1.1. respectively – lines 359-366).
Section 4.1: The authors mention the association between NETs and thromboembolic events. It would be valuable to explore this further, particularly in the context of obesity and how this might influence clinical management.
Response: We would like to thank the reviewer for this remark. We cite data from D’Abbondanza et al. (2019) that patients whose NET levels increased after bariatric surgery-induced weight reduction had thromboembolic events in anamneses (4/33 vs. 0/40). However, the authors do not explore this finding further, either retrospectively or prospectively.
The conclusion could be strengthened by summarizing the key takeaways and emphasizing the need for further research, especially regarding the therapeutic targeting of NETosis.
Response: As suggested by the reviewer, we tried our best to address the need for further research and potential new therapeutic approaches (section 8).
On the whole, the author lacks the necessary diagrams to help readers understand the content of the review more clearly.
Response: By the reviewer's suggestion, schematic representations (Figure 1-section 2.2. and Figure 2 – section 7.3.) were added to the manuscript.